# MHLDet: A Multi-Scale and High-Precision Lightweight Object Detector Based on Large Receptive Field and Attention Mechanism for Remote Sensing Images

Liming Zhou [1,2], Hang Zhao [1,2], Zhehao Liu [1,2], Kun Cai [1,2,*], Yang Liu [1,2,3] and Xianyu Zuo [1,2]

1 Henan Key Laboratory of Big Data Analysis and Processing, Henan University, Kaifeng 475004, China; lmzhou@henu.edu.cn (L.Z.); bless0929@henu.edu.cn (H.Z.); liuzhehao@henu.edu.cn (Z.L.); sea@vip.henu.edu.cn (Y.L.); xianyu_zuo@henu.edu.cn (X.Z.)
2 School of Computer and Information Engineering, Henan University, Kaifeng 475004, China
3 Henan Province Engineering Research Center of Spatial Information Processing and Shenzhen Research Institute, Henan University, Kaifeng 475004, China
* Correspondence: ckhenu@vip.henu.edu.cn

**Abstract:** Object detection in remote sensing images (RSIs) has become crucial in recent years. However, researchers often prioritize detecting small objects, neglecting medium- to large-sized ones. Moreover, detecting objects hidden in shadows is challenging. Additionally, most detectors have extensive parameters, leading to higher hardware costs. To address these issues, this paper proposes a multi-scale and high-precision lightweight object detector named MHLDet. Firstly, we integrated the SimAM attention mechanism into the backbone and constructed a new feature-extraction module called validity-neat feature extract (VNFE). This module captures more feature information while simultaneously reducing the number of parameters. Secondly, we propose an improved spatial pyramid pooling model, named SPPE, to integrate multi-scale feature information better, enhancing the model to detect multi-scale objects. Finally, this paper introduces the convolution aggregation crosslayer (CACL) into the network. This module can reduce the size of the feature map and enhance the ability to fuse context information, thereby obtaining a feature map with more semantic information. We performed evaluation experiments on both the SIMD dataset and the UCAS-AOD dataset. Compared to other methods, our approach achieved the highest detection accuracy. Furthermore, it reduced the number of parameters by 12.7% compared to YOLOv7-Tiny. The experimental results illustrated that our proposed method is more lightweight and exhibits superior detection accuracy compared to other lightweight models.

**Keywords:** object detection; remote sensing images (RSIs); lightweight network; attention mechanism; spatial pyramid pooling



## 1. Introduction

Object detection plays a crucial role in remote sensing image processing, enabling the automated extraction of objects of interest from RSIs, including buildings, roads, vehicles, and more. In the past, manual visual interpretation was the predominant method for acquiring geographic information. However, with the ongoing advancements in remote sensing technology and the continuous innovation of image-processing algorithms, object detection for RSIs has become one of the essential means to obtain large-scale geographic information efficiently. Currently, object detection technology in RSIs has found widespread applications across diverse domains, including urban planning [1], environmental monitoring [2], agricultural production [3], and other fields. This technology holds promising application prospects.

Traditional object-detection methods, such as scale-invariant feature transform (SIFT) [4], support vector machine (SVM) [5], and histogram of oriented gradients (HOG) [6], have

certain limitations due to manual features extraction and classifier construction. With the development of deep learning, object-detection methods can automatically learn features and construct efficient classifiers, resulting in higher accuracy and faster processing speeds. For example, Edge YOLO [7] can run in real-time on edge devices while maintaining high accuracy. At present, object detectors can be categorized into two main types: two-stage object detectors and one-stage object detectors. Two-stage detectors are represented by Faster R-CNN [8], Mask R-CNN [9], Cascade R-CNN [10], etc. Faster R-CNN is among the early pioneers of two-stage object detectors, distinguished in achieving both high detection accuracy and precise localization. The region proposal network (RPN) has been introduced to generate object candidate regions, followed by performing object classification and localization on these candidate regions. Mask R-CNN is a further advancement built upon the foundation of Faster R-CNN, which is able to not only detect object location and category, but also to generate accurate segmentation masks of objects. This makes Mask R-CNN perform well in image segmentation tasks. Cascade R-CNN is a two-stage detector proposed for small object detection, it significantly enhances the detection accuracy of small objects by means of cascade classifiers. Compared with other two-stage object detectors, Cascade R-CNN has higher detection accuracy for small objects, but slightly decreased detection accuracy for large objects.

Although two-stage detectors excel in high detection accuracy and positioning accuracy, their calculation speed is relatively slow. Moreover, training a two-stage object-detection model requires substantial data, and its efficiency diminishes when faced with a high volume of objects to detect. This situation is a challenge for both data acquisition cost and time cost. One-stage object detection models represented by SSD [11], RetinaNet [12], and you only look once (YOLO) [13] exhibit the advantages of high speed and applicability to scenarios such as real-time detection. With the continuous development of algorithms, their detection accuracy is gradually improved. However, the disadvantages of the one-stage detection model are low positioning accuracy, insensitivity to complex backgrounds, and occlusion, and these problems may have a certain impact on the target-detection effect of remote sensing images. In 2016, YOLOv1 [13] was proposed by Redmon et al., renowned for its fast detection speed, but the positioning accuracy of object detection is low. In 2017, Redmon et al. introduced YOLOv2 [14], which used DarkNet-19 as its backbone architecture. Despite achieving higher detection accuracy, it is still not friendly to small target recognition. In 2018, Redmon et al. introduced YOLOv3 [15], which used DarkNet-53 with the residual network to extract features from images. In 2019, Bochkovskiy proposed YOLOv4 [16]; he embedded the cross-stage partial (CSP) module into DarkNet-53, improving the accuracy and speed. In 2020, YOLOv5 was proposed with the Focus architecture, and it speeded up the training and improved the detection accuracy. In 2022, YOLOv7 [17] was proposed by Wang et al., which maintained a fast running speed and small memory consumption and achieved ideal results in detection accuracy. Although the YOLO series models have become some of the representative algorithms for object detection, single-stage object-detection methods still have certain limitations in object detection on large images such as RSIs.

In recent years, researchers have achieved notable advancements in the detection of small [18] or rotating object detection [19]. Yang et al. [20] proposed the sampling fusion network (SF-Net) to optimize for dense and small targets. Fu et al. [21] constructed a semantic module and a spatial layout module and then fed both into a contextual reasoning module to integrate contextual information about objects and their relationships. Han et al. [22] proposed the rotation-equivariant detector (ReDet), which explicitly codes rotational equivalence and rotational invariance to predict the direction accurately. Li et al. [23] proposed a model named OrientedRepPoints and a novel adaptive point learning quality assessment and sample allocation method, and they achieved more-accurate performance by accurately predicting the direction. However, while many researchers concentrate on challenges such as small object detection or the detection of rotating objects, they often overlook other common issues in RSIs' object detection. These issues include enhancing the detection of

multi-scale objects, tackling the difficulties of identifying objects concealed in shadows or with colors resembling the ground, and detecting objects with only partial features exposed in RSIs, as illustrated in Figure 1.

Considering the multi-faceted nature and complexity of current RSI problems, this paper proposes an enhanced detector, MHLDet, so as to address the issues mentioned above. Firstly, the VNFE is designed to alleviate the surroundings' impact and obtain more-robust feature information. In addition, it can concentrate more effectively on crucial regions to address the issue of partial feature detection. Secondly, we propose the SPPE with large receptive fields to fuse better multi-scale features to enhance the network's ability to detect targets of various scales. Finally, we introduce the CACL to preserve more feature information by minimizing the information loss during downsampling. Therefore, this model holds an advantage in detecting objects with colors close to the ground and in detecting multi-scale targets. In addition, it achieves higher precision and is more lightweight than existing methods. The primary contributions of this paper are outlined as follows:

1. This paper proposes a novel method for object detection in RSIs, which achieves a preferable balance between lightweight and accuracy.
2. We propose the VNFE module for more-effective feature extraction and integrated the SimAM attention mechanism into the backbone to focus on the key features. The VNFE module has fewer parameters, so it can reduce the computational cost more effectively and make the model lighter. In addition, the VNFE module can also enhance the generalization ability of the detector, as it can reduce the risk of overfitting while maintaining high accuracy.
3. This paper introduces a novel SPPE module by optimizing the SPPCSPC of YOLOv7-Tiny [17]. The SPPE is designed to enhance detection accuracy for medium- to large-sized objects while preserving the original advantages. As a result, it effectively fulfills the demands for detecting objects of various scales and addresses the application requirements.
4. The CACL module is designed to minimize the loss of feature information during the downsampling process in the neck network. This module can focus on feature regions with higher discrimination by performing attention calculations on the feature map and enriching the semantic information by fusing feature elements. Consequently, it improves the accuracy and robustness of the model.

The remainder of this study is structured as follows: Section 2 reviews related work on lightweight object detection algorithms and outlines the details of the proposed model. Section 3 introduces the implementation details of the experiments conducted on the SIMD and the UCAS-AOD datasets, and we present the results obtained from the ablation experiments. Section 4 carries out the discussion. Finally, Section 5 provides the conclusions from the study.

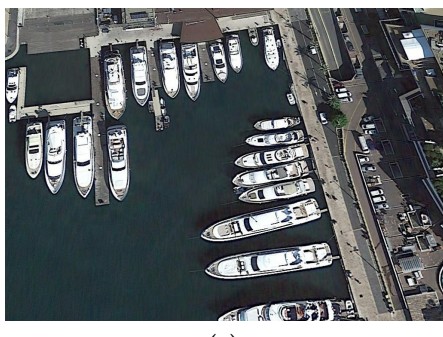 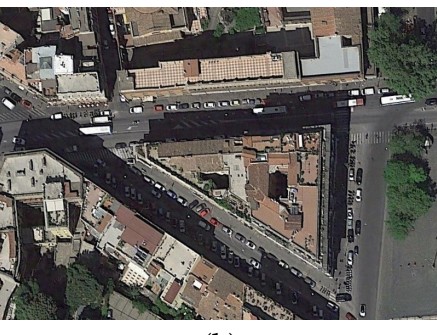 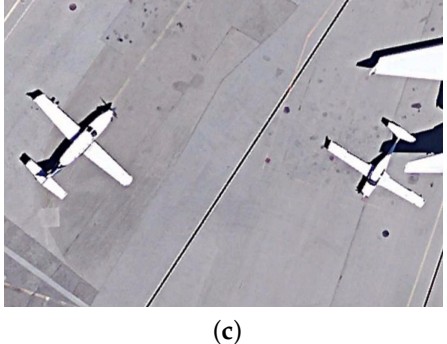

(**a**)　　　　　　　　　　　　　　(**b**)　　　　　　　　　　　　　　(**c**)

**Figure 1.** The object-detection issues in remote sensing images from the UCAS-AOD and the SIMD datasets. (**a**) represents the multi-scale objects of RSIs; (**b**) represents the objects hidden in shadow in RSIs; (**c**) represents the objects with only partial features exposed in RSIs.

## 2. Materials and Methods

### 2.1. Related Work

2.1.1. Lightweight Object-Detection Algorithms

In recent years, the one-stage detectors have garnered significant attention due to their fast execution speed and fewer parameters than two-stage detectors. YOLT [24] excels in multi-scale detection, precise localization, and real-time performance, making it particularly suitable for applications requiring high-quality detection of small objects. FCOS [25] regresses the bounding boxes at each position on the feature map to their corresponding positions on the original image. DETR [26] can predict the positions and categories of all objects in an image simultaneously. Zhu et al. proposed Deformable DETR [27], which achieves higher accuracy while reducing the training time. EfficientDet [28] introduced the bi-directional feature pyramid network (BiFPN) to achieve efficient multi-scale feature fusion. VFNet [29] introduced varifocal loss and an IoU-aware classification score to enhance the detection accuracy of dense objects. Aiming to obtain more-accurate features and enhance detection performance, R3DET [30] introduces the feature refinement module to address the misalignment between feature regions and feature maps in one-stage detectors. Moreover, it reduces the overall training memory consumption by approximately two-times.

Based on one-stage detectors, numerous scholars have proposed lightweight models. ShuffleNetV2 [31] stands out for its channel shuffling and efficient group convolutions, offering low computational costs for real-time applications. MobileNetV3 [32] introduces architectural optimizations, achieving improved accuracy while maintaining efficiency. YOLOv3-Tiny [15] is the earliest lightweight model in the YOLO series. By designing a simpler and more-efficient network structure, it achieves faster inference speed with fewer parameters in the YOLO series. With the proposal of the YOLOv7-Tiny network, it has reached a new peak in lightweight object-detection methods. However, there is still scope for further improvement in terms of both the detection accuracy and parameter optimization. Therefore, this paper proposes a new lightweight object-detection method aimed at improving the detection accuracy of medium- to large-sized objects and objects with no obvious features.

2.1.2. Attention Mechanism

The visual attention mechanism is a specialized process within human vision, aiding in swiftly identifying crucial information amidst cluttered surroundings. It establishes the focal point of attention, thus enhancing the efficient processing of visual information. The attention mechanism was applied to computer vision as early as 1998 by Laurent Itti [33], inspired by the neurons of the primate visual system, to merge multi-scale feature maps and select focal regions in order of saliency. In 2014, Google's DeepMind first applied the method of the attention mechanism on the recurrent neural network (RNN) model for image classification tasks, which attracted more people's attention [34].

In the field of computer vision, attention mechanisms can be categorized into two types: channel attention and spatial attention. Channel attention is represented by the squeeze-and-excitation network (SENet) model [35]. SENet initially compresses the feature map's spatial dimensions, subsequently employing network model learning to ascertain the significance of individual feature channels autonomously. This approach can assign varying weight coefficients to each channel, amplifying critical features while attenuating less-relevant ones. Spatial attention, such as spatial transformer networks (STNs) [36], can enable the transformation of data with different spatial deformations. It has the ability to automatically capture important regional features by adjusting and aligning the input data in a spatially adaptive manner.

However, it is essential to note that channel attention is a one-dimensional attention mechanism, which treats different channels differently, but treats all spatial locations equally. Spatial attention is a two-dimensional attention mechanism that treats different locations within the feature map differently while giving equal importance to all chan-

nels. SimAM [37] is a 3D parameter-free attention mechanism that enhances the feature extraction capability of neural networks. SimAM attaches 3D attention weights to feature maps, unlike traditional one-dimensional or two-dimensional weight attention methods. By considering the interdependencies among channels, spatial locations, and different depth levels, SimAM captures more-comprehensive and fine-grained information from the feature maps. This allows the network to focus on relevant features and improve its ability to extract meaningful representations, enhancing performance in various computer vision tasks.

### 2.1.3. Partial Convolution

Convolution is a common operation in object detection, extracting feature information by accessing all input channels. Multiple convolutional layers are employed for dimension transformation and feature extraction. However, this approach may encounter computation bottlenecks when dealing with huge input images or using larger convolutional kernels, augmenting the model's complexity. This solution leads to an increase in the number of parameters and computational requirements. For a feature map with the input $I \in R^{C \times H \times W}$, using $c$ filters $F \in R^{K \times K}$, the FLOPs would be:

$$h \times w \times k^2 \times c^2 \tag{1}$$

The partial convolution (PConv) mentioned in FastNet [38] exploits the redundancy in the feature map and systematically applies regular convolution (Conv) on only a portion of the input channels without affecting the remaining channels to maintain high FLOPS at reduced FLOPs, as a way to reduce latency and, thus, the parameters and computational effort of the model. The formula for calculating latency is as follows:

$$Latency = \frac{FLOPs}{FLOPS} \tag{2}$$

The FLOPs of PConv are:

$$h \times w \times k^2 \times c_p^2 \tag{3}$$

where $c_p = c/4$, and the FLOPs of PConv are only $1/16$ of the conventional convolution. Therefore, in the proposed VNFE module, we utilized PConv to effectively reduce the number of model parameters.

### 2.2. Proposed Method

The overall architecture of our MHLDet is depicted in Figure 2. In the backbone network, we embedded the VNFE module and SimAM to enhance the feature extraction ability while reducing the model's parameters. We introduced the SPPE module within the neck network that executes the pyramid pooling operations on the feature maps, which are obtained from the backbone network. This process aims to acquire multi-scale feature representations. The CACL module has been proposed to enhance feature expression capabilities and reduce information loss. Table 1 summarizes the main parameters of the proposed method.

In the backbone, $Input \in R^{1024 \times 1024 \times 3}$ represents the input images. It is downsampled by two CBL modules (consisting of convolutional layers, batch normalization, and the LeakyRELU activation function [39]); the convolution kernel size is $3 \times 3$, and the stride is set to 2. Moreover, the SimAM attention mechanism is embedded after max pool to help the network focus on key features, which can better represent important information in the input images to enhance the performance and the expression ability of the model. Then, we used the VNFE module to make the feature mapping improved, accelerate the training and convergence speed, and make the model more lightweight. The SPPE module is used to extract feature vectors of objects with multiple scales and expand their receptive fields. Feature maps of size $128 \times 128$, $64 \times 64$, and $32 \times 32$ are extracted from the backbone P3, P4, and P5 layers, respectively, and sent to the neck network, where feature fusion is performed.

As we all know, low-level features encompass local information such as edges and textures of objects, providing more spatial information, but limited semantic information. On the other hand, high-level features tend to possess richer semantic information. To address the loss of feature information due to downsampling and to make the neural network more robust and accurate for recognition, the features of the P3, P4, and P5 layers are fused by the CACL module.

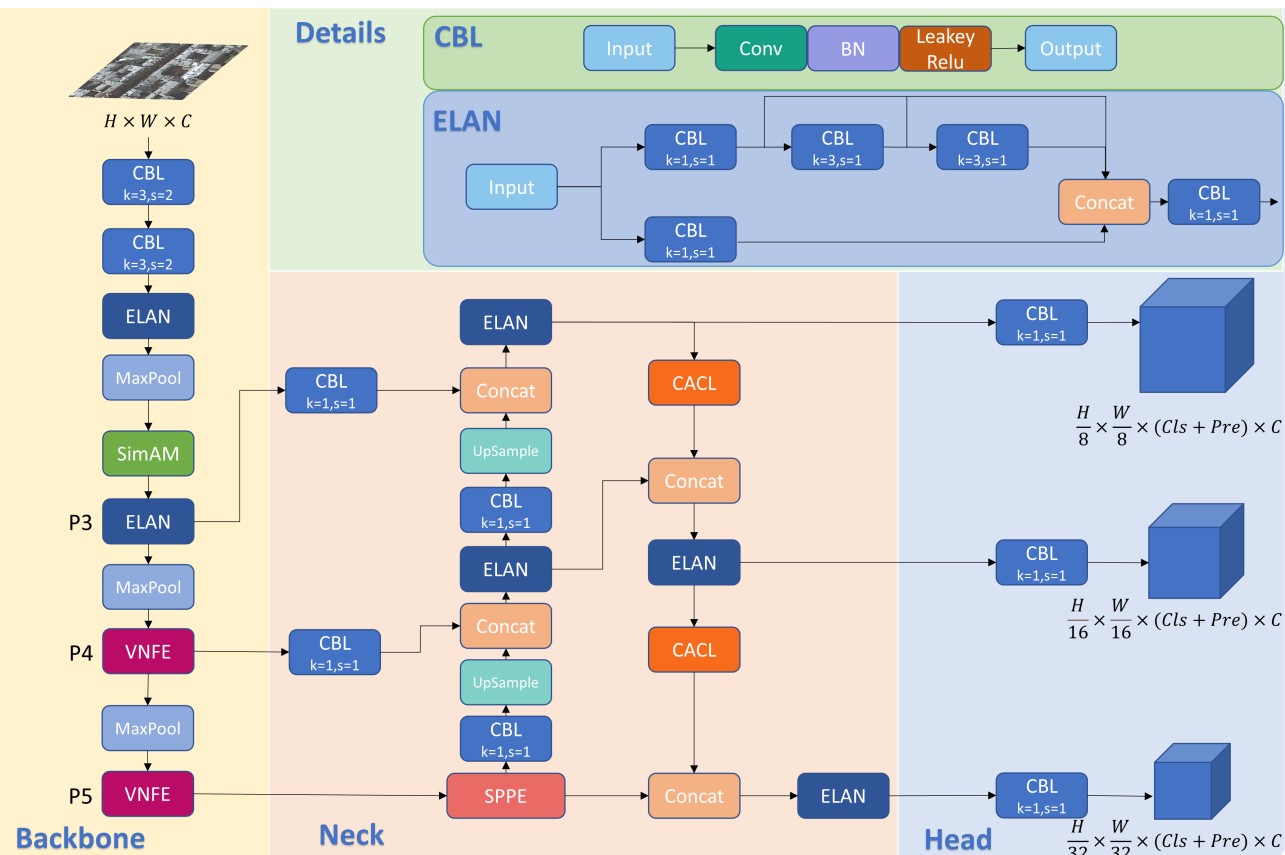

**Figure 2.** The overall structure of the MHLDet. We embedded the SimAM attention mechanism and VNFE module in the backbone and used the SPPE and CACL structure in the neck network. H, W, and C represent height, width, and channels, respectively. Cls represents the number of class predictions. Pre presents the bounding box predictions.

**Table 1.** Parameters of MHLDet.

| Number | Module | From | N | Output | Params |
|---|---|---|---|---|---|
| | Input | | | $1024 \times 1024 \times 3$ | |
| 0 | CBL | $-1$ | 1 | $512 \times 512 \times 32$ | 928 |
| 1 | CBL | $-1$ | 1 | $256 \times 256 \times 64$ | 18,560 |
| 2 | ELAN | $-1$ | 1 | $256 \times 256 \times 128$ | 31,104 |
| 3 | max pool | $-1$ | 1 | $128 \times 128 \times 128$ | 0 |
| 4 | SimAM | $-1$ | 1 | $128 \times 128 \times 64$ | 0 |
| 5 | ELAN | $-1$ | 1 | $128 \times 128 \times 128$ | 115,456 |
| 6 | max pool | $-1$ | 1 | $64 \times 64 \times 128$ | 0 |
| 7 | VNFE | $-1$ | 1 | $64 \times 64 \times 256$ | 242,432 |
| 8 | max pool | $-1$ | 1 | $32 \times 32 \times 256$ | 0 |
| 9 | VNFE | $-1$ | 1 | $32 \times 32 \times 512$ | 966,144 |
| 10 | SPPE | $-1$ | 1 | $32 \times 32 \times 256$ | 936,448 |
| 11 | CBL | $-1$ | 1 | $32 \times 32 \times 128$ | 33,024 |
| 12 | Upsample | $-1$ | 1 | $64 \times 64 \times 128$ | 0 |
| 13 | CBL | 7 | 1 | $64 \times 64 \times 128$ | 33,024 |

**Table 1.** *Cont.*

| Number | Module | From | N | Output | Params |
|:------:|:------:|:----:|:-:|:------:|:------:|
| 14 | Concat | (−1,12) | 1 | 64 × 64 × 256 | 0 |
| 15 | ELAN | −1 | 1 | 64 × 64 × 128 | 140,032 |
| 16 | CBL | −1 | 1 | 64 × 64 × 64 | 8320 |
| 17 | Upsample | −1 | 1 | 128 × 128 × 64 | 0 |
| 18 | CBL | 5 | 1 | 128 × 128 × 64 | 8320 |
| 19 | Concat | (−1,17) | 1 | 128 × 128 × 128 | 0 |
| 20 | ELAN | −1 | 1 | 128 × 128 × 64 | 35,200 |
| 21 | CACL | −1 | 1 | 64 × 64 × 128 | 82,432 |
| 22 | Concat | (−1,15) | 1 | 64 × 64 × 256 | 0 |
| 23 | ELAN | −1 | 1 | 64 × 64 × 128 | 140,032 |
| 24 | CACL | −1 | 1 | 32 × 32 × 256 | 328,704 |
| 25 | Concat | (−1,10) | 1 | 32 × 32 × 512 | 0 |
| 26 | ELAN | −1 | 1 | 32 × 32 × 256 | 558,592 |
| 27 | CBL | 20 | 1 | 128 × 128 × 128 | 73,984 |
| 28 | CBL | 23 | 1 | 64 × 64 × 256 | 295,424 |
| 29 | CBL | 26 | 1 | 32 × 32 × 512 | 1,180,672 |
| 30 | Detect | | 1 | 3 | 55,016 |
| | Total | | | | 5,283,848 |

### 2.2.1. Validity-Neat Feature Extract Module

For the original image input to the model, excessive convolutional and pooling layers in the backbone network will not only lead to feature redundancy but also lead to computational redundancy. We propose using the VNFE module to replace the regular convolutional layers of the original backbone network.

As shown in Figure 3, the PBL module plays a role in downsampling the feature map and extracting high-level features. Specifically, the PConv within this module extracts spatial features from the input channels, leaving the other channels unaltered. This strategic approach reduces redundant computations and minimizes memory usage, contributing to the overall lightweight nature of the model. Furthermore, SimAM is integrated into the second branch of the VNFE module. It aims to assist the network in emphasizing key features and appending crucial information onto the feature maps without introducing extra parameters. The formula is as follows:

$$Y_1 = P_{3,1}(P_{3,1}(F_{1,1}(X))) \tag{4}$$

$$Y_2 = P_{3,1}((F_{1,1}(X)) + (SimAM(X))) \tag{5}$$

$$Y_3 = F_{1,1}(Y_1 \oplus Y_2) \tag{6}$$

where $X$ represents the feature input, $F_{1,1}$ denotes a convolution operation with a filter size $1 \times 1$ and stride of 1, $P_{3,1}$ is a PConv operation with a filter size of $3 \times 3$ and stride of 1, SimAM represents the attention operation, and $\oplus$ is the concatenate operation.

SimAM attention distinguishes the importance of neurons by defining an energy function for each neuron. Specifically, an energy function is defined for each neuron using the following formula:

$$e_t(w_t, b_t, y, x_i) = \frac{1}{M-1} \sum_{i=1}^{M-1} (-1 - (\omega_t x_i + b_t))^2 \\ + (1 - (\omega_t t + b_t))^2 + \lambda \omega_t^2 \tag{7}$$

where $t$ denotes the target neuron while $x_i$ denotes the other neurons in a single channel of the input feature $X \in R^{(C \times H \times W)}$, $i$ is the index in the spatial dimension, and $M = H \times W$

is the number of all neurons in a single channel. $\omega_t$ and $b_t$ are the weights and biases of the transformation, which can be expressed as follows:

$$\omega_t = -\frac{2(t - \mu_t)}{(t - \mu_t)^2 + 2\sigma_t^2 + 2\lambda} \tag{8}$$

$$b_t = -\frac{1}{2}(t + \mu_t)\omega_t \tag{9}$$

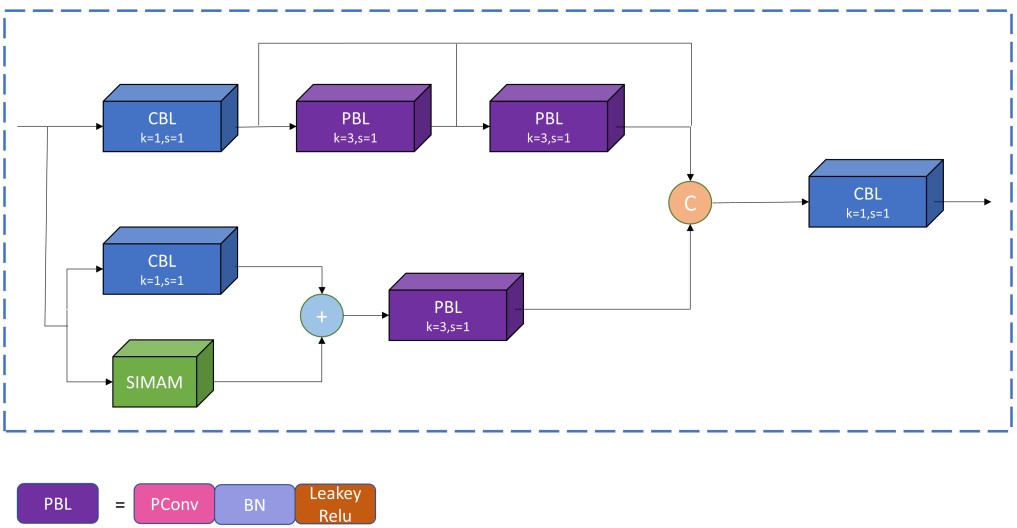

**Figure 3.** Structure of the VNFE module.

$\mu_t$ and $\sigma_t^2$ are the mean and variance of all neurons in this channel, except $t$:

$$\mu_t = \frac{1}{M-1}\sum_{i=1}^{M-1} x_i \tag{10}$$

$$\sigma_t^2 = \frac{1}{M-1}\sum_{i=1}^{M-1}(x_i - \mu_t)^2 \tag{11}$$

Assuming the same distribution among all the pixels in a single channel, we can calculate the mean and variance of all the neurons, which allows us to derive the minimum energy formula:

$$e_t^* = \frac{4(\hat{\sigma}^2 + \lambda)}{(t - \hat{\mu})^2 + 2\hat{\sigma}^2 + 2\lambda} \tag{12}$$

Equation (12) indicates that the energy value of $e_t^*$ reflects the significance of the contrast between neuron t and nearby neurons, and a lower energy value indicates a more-pronounced difference and higher importance of neuron $t$ in the context of nearby neurons. $\lambda$ is the regularization coefficient. The entire refinement phase of SimAM can be calculated as follows:

$$SimAM(X) = sigmoid(\frac{1}{E}) \times X \tag{13}$$

where $E$ groups all $e_t^*$ from the spatial and channel dimensions. Besides, we used sigmoid activation to constrain excessive $E$ values.

### 2.2.2. Spatial Pyramid Pooling Enforce Module

The current spatial pyramid pooling method still has potential for improvement when detecting medium- to large-sized targets. In order to better enhance the detection ability of the model for multi-scale targets, we learned from the SPPCSPC [17] module and propose

a new spatial pyramid pooling enforce (SPPE) module. In the first branch, multi-scale feature information is captured by constructing pyramid levels at different scales. Pooling operations at different scales enable the network to perceive and process target objects of different scales and sizes of the input image, thereby improving the adaptability and generalization ability of the model. The larger pooling operation can retain the overall features and spatial layout of the target object. By increasing the convolution kernel size, the model can cover more details and context information of the target area, which is helpful in detecting medium- to large-sized targets more accurately. Therefore, we changed the connection mode of the maximum pooling layers. The receptive fields increased to 5, 13, and 25, respectively, so as to enhance the detection capability for multi-scale objects. The receptive field can be calculated using the following formula:

$$
\begin{cases}
r_0 = 1, \\
r_1 = k_1, \\
r_n = r_{n-1} + (k_n - 1) \prod\limits_{i=1}^{n-1} s_i. \quad (n \geq 2)
\end{cases}
\tag{14}
$$

The input is passed through the CBL and PBL in the second branch, respectively. The number of output channels is reduced by half compared to the original without changing the feature map size. These channels are subsequently element by element multiplied to increase the amount of information in each dimension of the feature map, which can supplement the feature detail information to obtain a richer and more-comprehensive feature representation. In addition, it makes up for the attenuation of the small target detection ability due to the increase of the receptive field. Finally, the two branches are concatenated in the channel dimension. Then, the number of output channels of the SPPE was adjusted to be consistent with the input through a CBL with a $1 \times 1$ convolutional kernel size. The SPPE structure is illustrated in Figure 4.

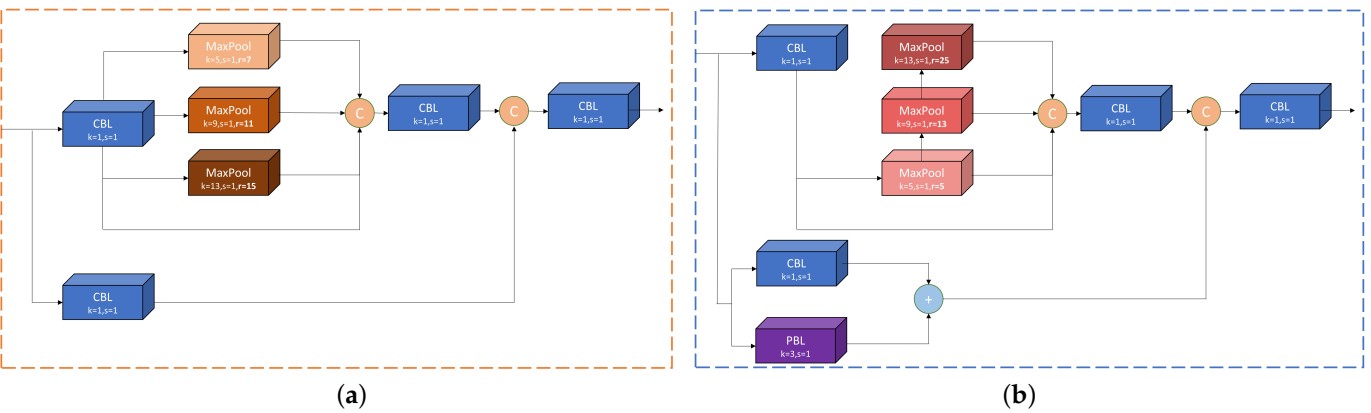

(**a**) (**b**)

**Figure 4.** Structure of the SPPCSPC and SPPE. (**a**) represents the SPPCSPC; (**b**) represents the SPPE.

### 2.2.3. Convolution Aggregation Cross Layer

Using a CBL with only a $3 \times 3$ convolution for downsampling can result in information loss and decrease feature quality. To minimize the information loss caused by downsampling, we embedded the CACL module within the network, allowing the feature map to retain more feature information. In the first branch of the CACL, compared with the max pooling layer, using the average pooling layer can better preserve the spatial information of the feature map and effectively utilize the global information. After that, a $1 \times 1$ convolution is used to deepen the network, introduce more nonlinearity, and enhance the neural network's expressive capability without expanding the receptive field. The feature map's channel count is doubled as well. Then, the network model focuses on the feature regions with higher discrimination by using the SimAM attention mechanism.

We used a CBL module with a convolution kernel size of $3 \times 3$ and a stride of 2 in the second branch to reduce the feature map size through downsampling. This downsampling process helps maintain crucial feature information while enhancing the model's robustness.

By adding the two branches, the module can fuse different feature expressions, improve the diversity of features and the expression ability, and enhance the representation ability and detection performance. Its structure is illustrated in Figure 5, and the formula is as follows:

$$O = SimAM(F_{1,1}(AvgPool(X))) + F_{3,2}(X) \tag{15}$$

where $O$ and $X$ represent the output and input feature maps of this module, SimAM represents the attention operation, $F_{3,2}$ represents a convolution operation with filter size $3 \times 3$ and a stride of 2, and AvgPool represents the average pooling operation.

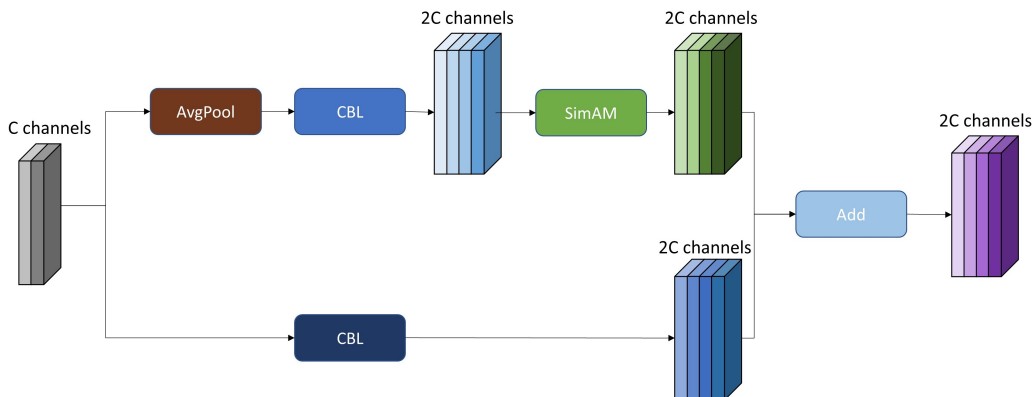

**Figure 5.** Structure of the CACL module.

### 3. Results

*3.1. Datasets*

We conducted the experiments using the SIMD [40] and UCAS-AOD [41] datasets to evaluate the effectiveness. Figure 6 shows some examples of the datasets this paper used.

The SIMD dataset is a recently released high-resolution remote sensing dataset. It is characterized by its high resolution and the inclusion of multi-scale imagery. This dataset contains 5000 high-resolution images with 15 categories, Car, Truck, Van, Long Vehicle, Bus, Airliner, Propeller Aircraft, Trainer Aircraft, Chartered Aircraft, Fighter Aircraft, Others, Stair Truck, Pushback Truck, Helicopter, Boat, respectively. There are a total of 45,096 instances, and the image size is $1024 \times 768$ pixels. We used 4000 images for training and 1000 images for validation according to the division of the original dataset.

The UCAS-AOD dataset is specifically designed for the detection of airplanes and cars. The image size is $1280 \times 659$ pixels, and it contains two categories of airplane and car. The objects in the images are evenly distributed in terms of their orientations. In this experiment, we removed the negative example images that did not contain any example images and only kept the images that contained the two classes of airplane and car. The training set, validation set, and test set of the UCAS-AOD dataset were randomly divided with a ratio of 7:2:1.

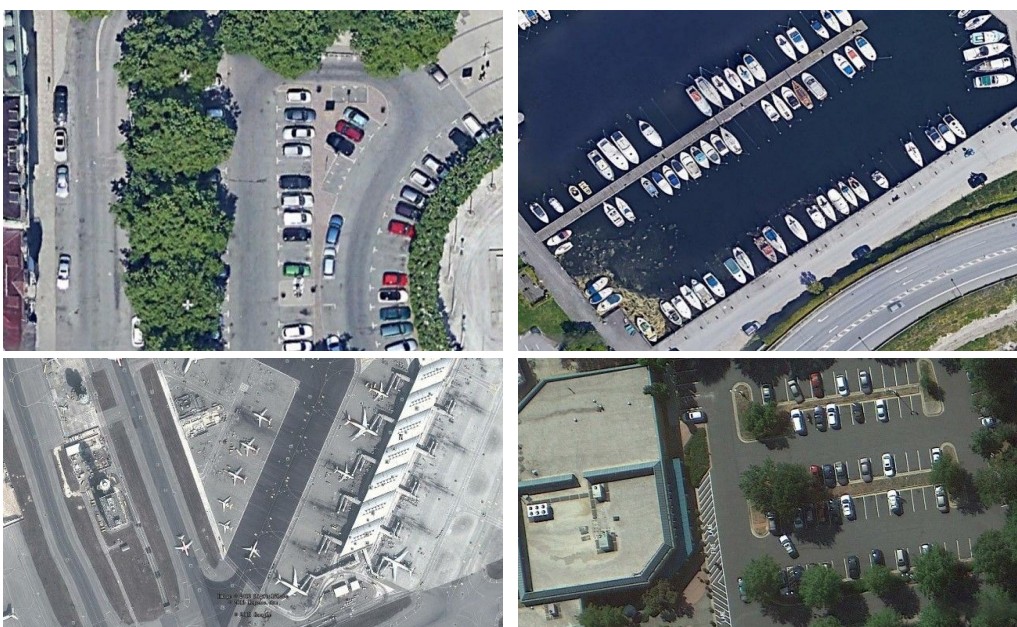

**Figure 6.** Some examples of the remote sensing images utilized for training and evaluating our method from the SIMD and UCAS-AOD datasets.

### 3.2. Evaluation Criteria

To evaluate the performance of our proposed detector, we used several widely used evaluation metrics. These metrics included the precision ($P$), recall ($R$), average precision ($AP$), and mean average precision ($mAP$). The formulas for precision and recall are as follows:

$$Precision = \frac{TP}{TP + FP} \tag{16}$$

$$Recall = \frac{TP}{TP + FN} \tag{17}$$

$$AP = \int_0^1 P(R)\mathrm{d}R \tag{18}$$

where $TP$ is true positive and is the number of positive samples that were correctly predicted to be positive. $FP$ is false positive, which is the number of false positives that were incorrectly predicted. $FN$ is false negative and is the number of positive samples that were incorrectly predicted as negative. $AP$ is the area under the precision–recall curve (P–R curve) and ranges from 0 to 1. $mAP$ is a commonly used comprehensive evaluation metric in object detection, which comprehensively considers the precision and recall of different categories. We calculated the area under the P–R curve for each class and, then, averaged the $AP$ across all classes to obtain the $mAP$. The $mAP$ provides an overall evaluation of model performance and can be used to compare the strengths and weaknesses of different models.

$$mAP = \frac{1}{n}\sum_{i=1}^{n} AP_i \tag{19}$$

where $n$ represents the total number of categories in the detection task.

The intersection over union ($IoU$), is a metric used to measure the degree of overlap between the predicted bounding box ($B_p$) and the true bounding box ($B_g t$) in the object-detection task. It is defined by computing the ratio between the intersection area

of the predicted bounding box and the true bounding box and the area of their union. The calculation formula is as follows:

$$IoU = \frac{area(B_p \cap B_{gt})}{area(B_p \cup B_{gt})} \tag{20}$$

In addition, we followed the COCO evaluation metrics [42], including the mAP ($IoU$ threshold = 0.5), $mAP_{0.75}$, and $mAP_{0.5:0.95}$, and it also includes $AP_s$ (for small objects, area < $32^2$), $AP_m$ (for medium objects, $32^2$ < area < $96^2$), and $AP_l$ (for large objects, $96^2$ < area).

These metrics can help us understand the efficiency of the model in practical applications. Combining these evaluation metrics, we can comprehensively assess the performance of our proposed method in regard to accuracy, efficiency, and practicality and compare it with current methods.

### 3.3. Parameter Setting

The experimental environment was configured as follows: the computer was equipped with an NVIDIA GeForce RTX 3070 graphics card (8GB); the CPU was an Intel Core i7-10700K; the operating system was Ubuntu 20.04.4 LTS. We used the PyTorch deep learning framework (Version 1.13.1) for model development and training in Python 3.8 and CUDA 11.4.

The input image size was 1024 × 1024; the batch size was four; the optimization was stochastic gradient descent (SGD); the momentum was 0.937; the weight decay was set to 0.0005. To improve the convergence and training efficiency of the model, we performed three epochs of warm-up training and set the warm-up momentum to 0.8 initially. The initial train parameters are shown in Table 2.

**Table 2.** The initialization training parameters on the SIMD dataset.

| Input Size | Batch Size | Momentum | Warm-Up Momentum | Weight Decay | Learning Rate |
|---|---|---|---|---|---|
| 1024 × 1024 | 4 | 0.937 | 0.8 | 0.0005 | 0.01 |

### 3.4. Experimental Results

3.4.1. Experimental Results on the SIMD Dataset

This paper took YOLOv7-Tiny [17] as the baseline and conducted experiments on the SIMD dataset to validate the effectiveness of our method and compare it with other one-stage detectors. The detectors for comparison were the anchor-free detector FCOS [25], TOOD [43], PP-Picodet-m [44], YOLOX-S [45], YOLOv6-N, YOLOv6-S [46], and YOLOv8-N [47], and the anchor-based detectors were RetinaNet [12], YOLOv5-N, YOLOv5-S [48], YOLO-HR-N [49], and YOLOv7-Tiny. We did not compare with two-stage detectors. In our experiments, we focused on proposing a lightweight model designed to meet the needs of scenarios with limited computational resources, and our approach focused more on reducing the complexity and computational overhead of the model while maintaining high performance compared to traditional two-stage models. Our experiments on the SIMD dataset used the K-Means++ [50] algorithm to generate a new anchor, and the experimental data after using the new anchor are shown in Table 3. In addition, to display the results more directly, this paper assigned an abbreviation name (AN) to each category in the dataset, as shown in Table 4.

**Table 3.** The results of YOLOv7-Tiny with and without the K-Means++ algorithm on the SIMD dataset.

| Algorithm | mAP (%) | $mAP_{0.5:0.95}$ (%) | Params (m) | FLOPs (G) |
|---|---|---|---|---|
| YOLOv7-Tiny | 82.3 | 64.2 | 6.05 | 13.3 |
| YOLOv7-Tiny+K-Means++ | 82.7 | 64.8 | 6.05 | 13.3 |

**Table 4.** Categories and their corresponding names on the SIMD dataset.

| Category | AN | Category | AN | Category | AN |
|---|---|---|---|---|---|
| Car | CR | Airliner | AL | Others | OT |
| Truck | TR | Propeller Aircraft | PA | Stair Truck | ST |
| Van | VAN | Trainer Aircraft | TA | Pushback Truck | PT |
| Long Vehicle | LV | Chartered Aircraft | CA | Helicopter | HC |
| Bus | BUS | Fighter Aircraft | FA | Boat | BO |

Table 5 shows the detection accuracy results of our detector and other detectors in various categories on the SIMD dataset. It is evident that our method achieved optimal accuracy across the majority of categories, with the mAP reaching 84.7%. As shown in the table, our mAP was 2.4% higher than YOLOv7-Tiny. Additionally, it was nearly 10% higher than YOLOv5-N.

**Table 5.** The detection accuracy of each category on the SIMD dataset.

| Model | CR | TR | VAN | LV | BUS | AL | PA | TA | CA | FA | OT | ST | PT | HC | BO | mAP |
|---|---|---|---|---|---|---|---|---|---|---|---|---|---|---|---|---|
| TOOD [43] | 63.7 | 58.9 | 57.2 | 56.7 | 67.9 | 83.0 | 78.4 | 78.5 | 78.9 | 76.5 | 10.9 | 24.1 | 15.2 | 30.8 | 75.5 | 74.3 |
| RetinaNet [12] | 66.5 | 54.6 | 57.9 | 52.9 | 62.3 | 84.7 | 72.6 | 76.6 | 77.0 | 84.6 | 7.2 | 21.6 | 5.3 | 33.4 | 73.3 | 71.7 |
| FCOS [25] | 69.5 | 58.2 | 61.7 | 52.5 | 59.9 | 79.1 | 70.7 | 73.9 | 72.1 | 82.5 | 16.2 | 26.2 | 9.1 | 42.0 | 72.5 | 75.9 |
| YOLOv5-N [48] | 94.2 | 83.2 | 84.5 | 83.2 | 91.8 | 96.3 | 85.2 | 95.2 | 92.9 | 88.8 | 23.1 | 44.6 | 15.5 | 58.3 | 97.9 | 75.7 |
| YOLOv5-S [48] | 94.4 | 85.2 | 86.7 | 83.4 | 92.7 | 96.4 | 94.8 | 96.5 | **96.3** | 92.6 | **33.8** | 54.2 | 47.7 | 88.9 | 98.4 | 82.8 |
| YOLOv7-Tiny [17] | 94.5 | 85.6 | 86.2 | 86.3 | 93.7 | 97.6 | **96.6** | 95.9 | 94.2 | **99.5** | 28.1 | 50.5 | 48.3 | 78.2 | 98.6 | 82.3 |
| Ours | **94.7** | **85.7** | **87.5** | **87.2** | **94.7** | **97.6** | 96.1 | **97.8** | 95.8 | 94.6 | 30.1 | **54.5** | **58.8** | **96.8** | **98.9** | **84.7** |

Bold text is the optimal value for the column.

There are 15 categories in the SIMD dataset, and our detection efficiency in most categories reached an optimal level. In dense scenes, such as Car (CR) and Long Vehicle (LV), our MHLDet achieved the best performance. At the same time, it had a significant improvement compared with other models on medium- to large-sized targets such as Airliner (ALR), Chartered Aircraft (CA), and Fighter Aircraft (FA). The observed improvement can be attributed to the proposed VNFE module and the embedded SimAM attention mechanism, which effectively leverages the feature information within the backbone and enhances the feature representation, while the inclusion of the SPPE module expands the receptive fields, making it more conducive for detecting medium- to large-sized objects, and the CACL module reduced the feature loss in the process of downsampling. The experimental results further validated the effectiveness and robustness of the proposed method. Table 6 shows the performance comparison of MHLDet with other one-stage methods.

On the SIMD dataset, our MHLDet detector achieved an 84.7% mAP with 5.28 m model parameters. As shown in Figure 7, the proposed method outperformed all compared

one-stage methods, achieving the highest detection accuracy with fewer parameters. This excellence enabled our model to excel in resource-constrained environments. Compared with the baseline, the parameters of the model proposed by us decreased by 12.7%, and the mAP increased by 2.4%; the $mAP_{0.75}$ and $mAP_{0.5:0.95}$ also increased by 2.9% and 2.6%, respectively. On medium- to large-sized objects, $AP_m$ and $AP_l$ increased by 2.4% and 1.3%, respectively, compared to the baseline. In particular, the proposed model achieved the highest performance scores the mAP, $mAP_{0.75}$, $mAP_{0.5:0.95}$, and small-, medium-, and large-scale AP values, while maintaining fewer parameters. Compared with the anchor-free detectors PP-PicoDet-m, YOLOv6-N, and YOLOv8-N, these detectors' parameters were slightly fewer than MHLDet, but our mAP showed an increase of 11.8%, 10.3%, and 3.1%. In comparison to the anchor-based detectors, YOLOv5-N had the fewest parameters, but we were 9% and 8.9% higher in the mAP and $mAP_{0.5:0.95}$, respectively. In the aspect of multi-scale object detection, our $AP_m$ and $AP_l$ were 12.3% and 10.1% higher than their counterparts.

**Table 6.** Comparison with state-of-the-art lightweight detectors on the SIMD dataset.

| Method | mAP | $mAP_{0.75}$ (%) | $mAP_{0.5:0.95}$ (%) | $AP_s$ (%) | $AP_m$ (%) | $AP_l$ (%) | Params (m) | FLOPs (G) |
|---|---|---|---|---|---|---|---|---|
| *Anchor-free* | | | | | | | | |
| FCOS (2019) [25] | 75.9 | 67.2 | 56.4 | 5.9 | 47.4 | 61.2 | 31.87 | 201.9 |
| TOOD (2021) [43] | 74.3 | 68.8 | 57.1 | 12.3 | 49.4 | 63.3 | 31.98 | 188.7 |
| PP-PicoDet-m (2021) [44] | 72.9 | 67.7 | 56.7 | 15.7 | 50.3 | 61.6 | 3.43 | - |
| YOLOX-S (2021) [45] | 80.3 | 74.8 | 62.7 | 12.2 | 59.5 | 67.8 | 8.94 | 68.5 |
| YOLOv6-N (2022) [46] | 74.4 | 69.0 | 58.5 | 8.2 | 49.3 | 63.1 | 4.7 | 11.4 |
| YOLOv6-S (2022) [46] | 78.9 | 73.1 | 62.7 | 8.5 | 57.8 | 69.2 | 18.5 | 45.3 |
| YOLOv8-N (2023) [47] | 81.6 | - | 65.9 | - | - | - | 3.01 | 8.2 |
| *Anchor-based* | | | | | | | | |
| YOLO-HR-N (2022) [49] | 83.0 | - | 64.0 | - | - | - | 3.34 | 4.4 |
| RetinaNet (2017) [12] | 71.7 | 65.5 | 55.4 | 5.4 | 47.7 | 61.5 | 36.39 | 215.5 |
| YOLOv5-N (2020) [48] | 75.7 | 69.3 | 57.9 | 15.1 | 49.6 | 62.0 | 1.78 | 4.3 |
| YOLOv5-S (2020) [48] | 82.8 | 74.9 | 64.4 | 11.2 | 61.9 | 68.0 | 7.06 | 16.1 |
| YOLOv7-Tiny (2022) [17] | 82.3 | 75.4 | 64.2 | 19.3 | 59.5 | 70.8 | 6.05 | 13.3 |
| MHLDet (ours) | 84.7 | 78.3 | 66.8 | 20.1 | 61.9 | 72.1 | 5.28 | 12.2 |

By comparing the $AP_s$, $AP_m$, and $AP_l$ with each model, we found that the gains of our model mainly came from medium objects and large objects. Although the accuracy on medium objects was the same as that of YOLOv5-S, the accuracy on $AP_s$ and $AP_l$ improved by 8.9% and 4.1%, respectively. Despite the MHLDet model having 3.5 M more parameters compared to YOLOv5-N, it remains categorized as a lightweight model.

Figure 8 depicts the visual comparison of the heat map of the baseline with our proposed method, and it can be seen that our proposed method excels in predicting objects with only partial features and objects that are close to the surface color. To be specific, the heatmap's colors transition from blue to red on the images, symbolizing the probability of target presence increasing from low to high. Blue areas indicate the potential absence of targets, while red regions signify a high likelihood of target presence. Figure 8(b1–b3) are the results of YOLOv7-Tiny, and Figure 8(c1–c3) are the results of our proposed model. From Figure 8(b1,c1), it can be observed that we also had a heatmap representation on the ship just below, where only a portion of the features was exposed. As shown in Figure 8(b2,c2), YOLOv7-Tiny overlooked the middle vehicle, where our model correctly annotated it. From Figure 8(b3,c3), it is evident that our model provided more-accurate and continuous annotations for vehicles in shadow. As a result, our model can accurately depict the distribution of data. The visualization detection results are shown in Figure 9.

To visually illustrate the improvement achieved by our proposed model, this paper presents the detection results for some typical scenes. Figure 9(b1–b6) are the results of YOLOv7-Tiny, and Figure 9(c1–c6) are the results of our proposed model. From Figure 9(b1,c1), our model performed significantly better than YOLOv7-Tiny in terms of detecting small objects. From Figure 9(b2–b4,c2–c4), it can be observed that our model detected more

objects that closely matched the color of the ground. As shown in Figure 9(b5,c5), our model detected the object that exhibited only partial features. In addition, our model detected a higher number of densely arranged objects in Figure 9(b6,c6).

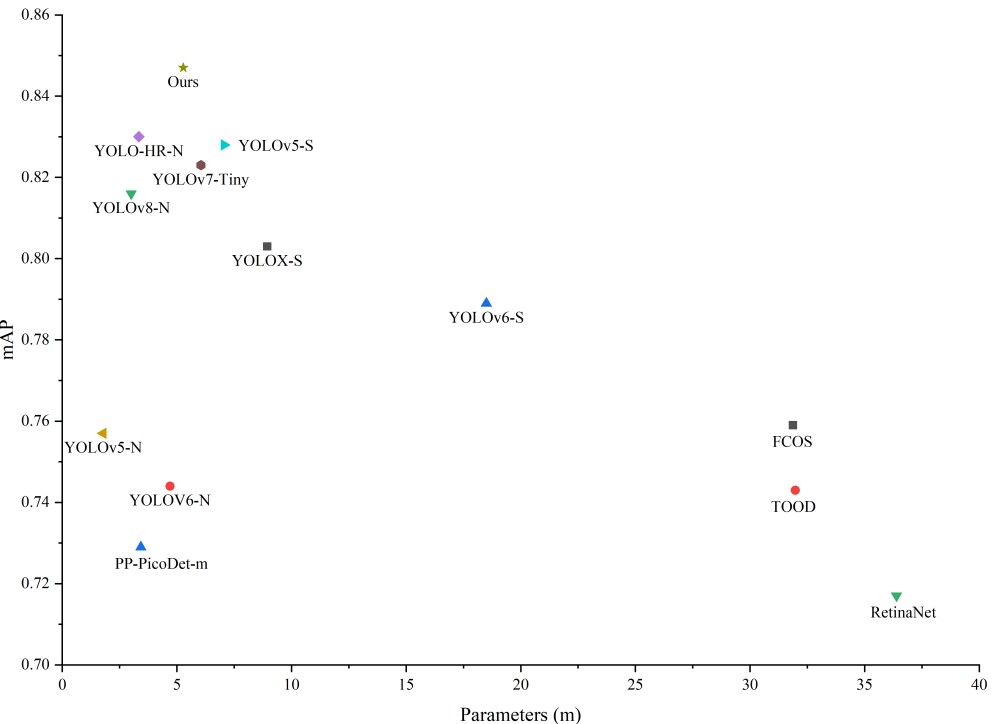

**Figure 7.** Comparison of the mAP and parameters of different detectors on the SIMD dataset.

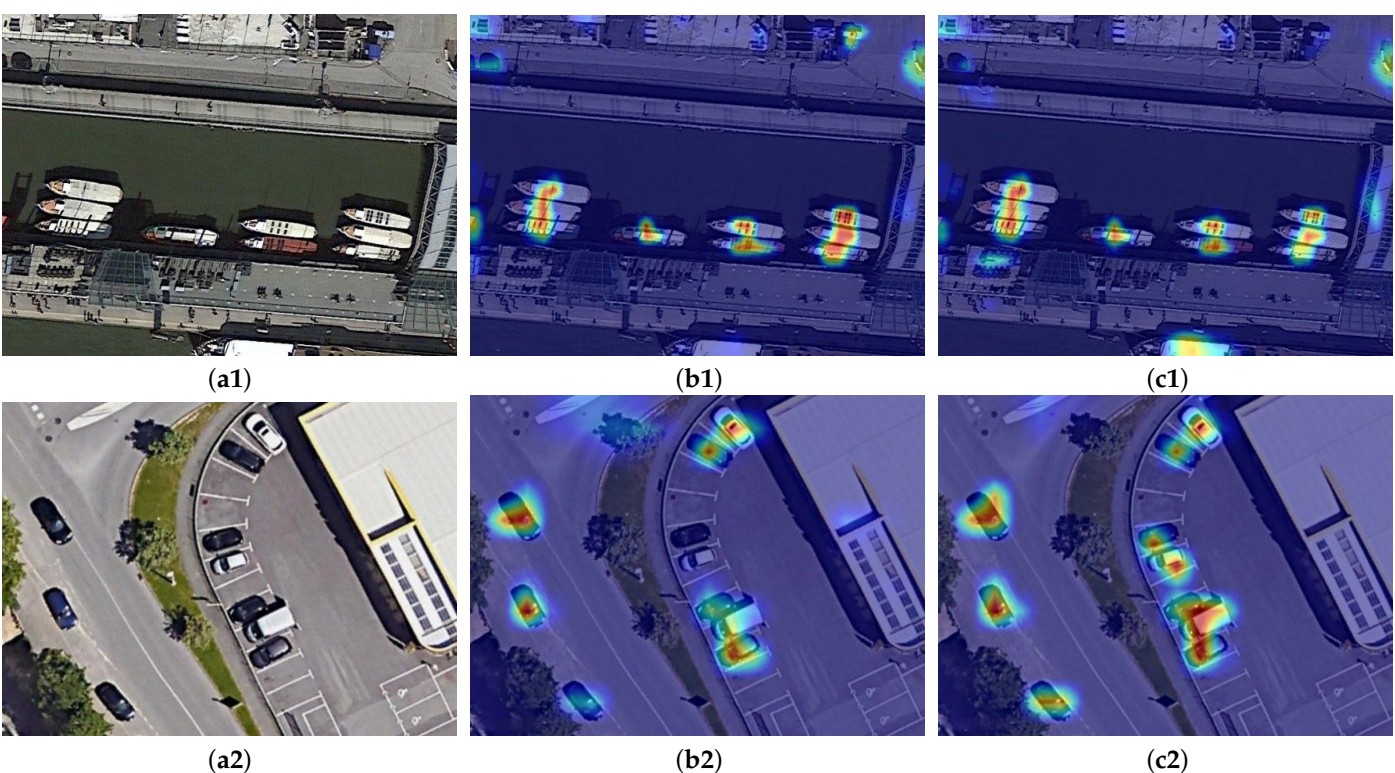

**Figure 8.** *Cont.*

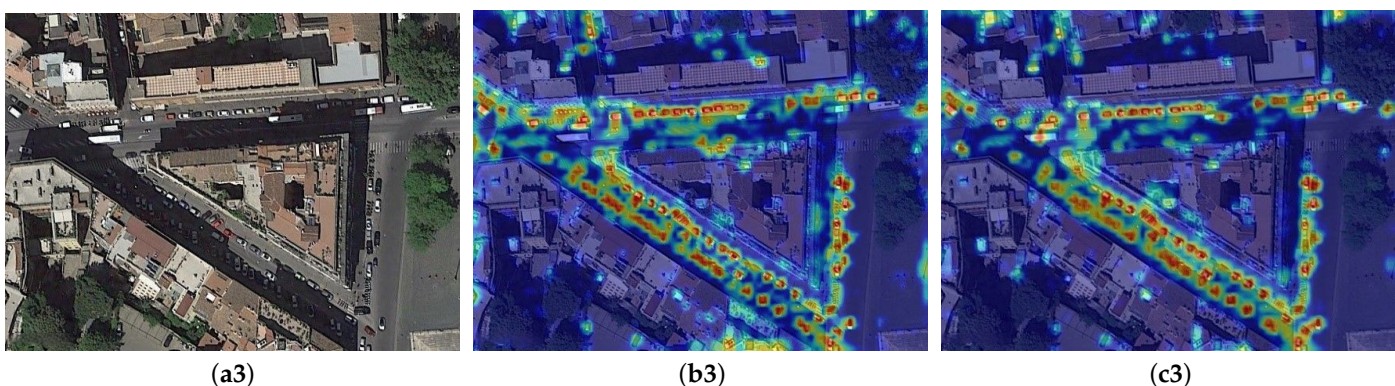

(**a3**)　　　　　　　　(**b3**)　　　　　　　　(**c3**)

**Figure 8.** Heatmap visualization results on the SIMD dataset. (**a1**–**a3**) represents the original images; (**b1**–**b3**) represents the heatmap visualization results of YOLOv7-Tiny; (**c1**–**c3**) represents the heatmap visualization results of the method we proposed.

(**a1**)　　　　　　　　(**b1**)　　　　　　　　(**c1**)

(**a2**)　　　　　　　　(**b2**)　　　　　　　　(**c2**)

(**a3**)　　　　　　　　(**b3**)　　　　　　　　(**c3**)

**Figure 9.** *Cont.*

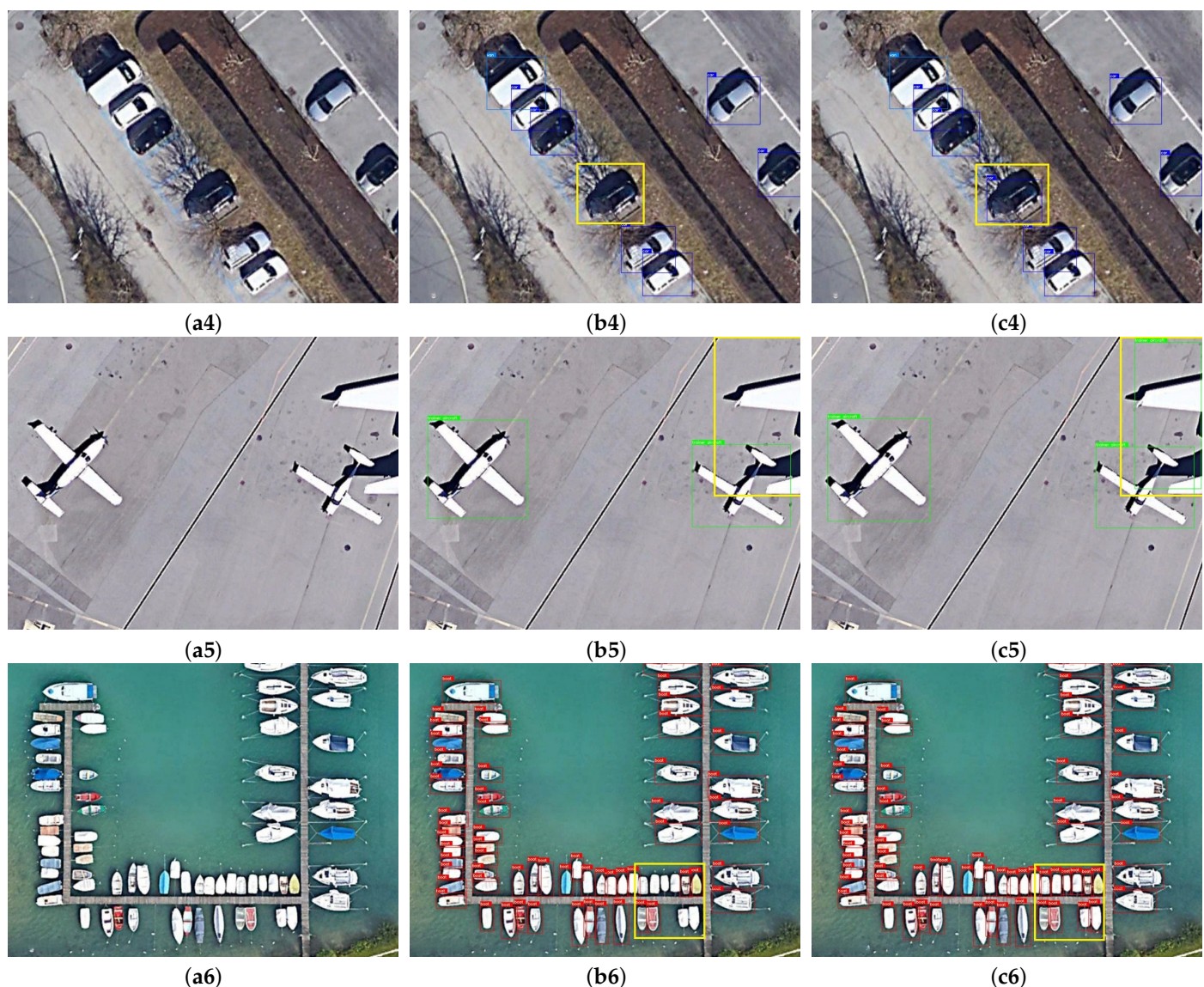

**Figure 9.** The visualization results of baseline and MHLDet on the SIMD dataset. (**a1–a6**) represents the original images; (**b1–b6**) represents the visualization results of YOLOv7-Tiny; (**c1–c6**) represents the visualization results of the method we proposed. The yellow boxes show the differences between YOLOv7-Tiny and MHLDet.

### 3.4.2. Experimental Results on the UCAS-AOD Dataset

Table 7 shows the accuracy comparison of our method with other methods on the UCAS-AOD dataset. It is evident from the table that our proposed method achieved the highest mAP among all the compared methods. Our method demonstrated a 4.6% improvement over the second-best method for small and 2% for large objects.

**Table 7.** The detection accuracy of different methods on the UCAS-AOD dataset.

| Method | $AP_s$ (%) | $AP_m$ (%) | $AP_l$ (%) | mAP |
|---|---|---|---|---|
| YOLOv5-N [48] | 17.2 | 54.5 | 50.1 | 95.9 |
| YOLOv6-N [46] | 11.4 | 53.8 | 44.2 | 93.63 |
| YOLOv6-S [46] | 9.5 | 57.4 | 49.1 | 94.9 |
| YOLOv7-Tiny [17] | 18.2 | 57.4 | 48.2 | 95.4 |
| MHLDet (Ours) | 23.8 | 57.6 | 52.1 | 96.4 |

Based on the data presented in Table 7, our proposed method improved in all scales compared to other methods, and it can be competent for the vast majority of object-detection tasks by the excellent performance compared to current state-of-the-art methods. Figure 10 shows the visualization results on the UCAS-AOD dataset, further supporting the effectiveness of our proposed method.

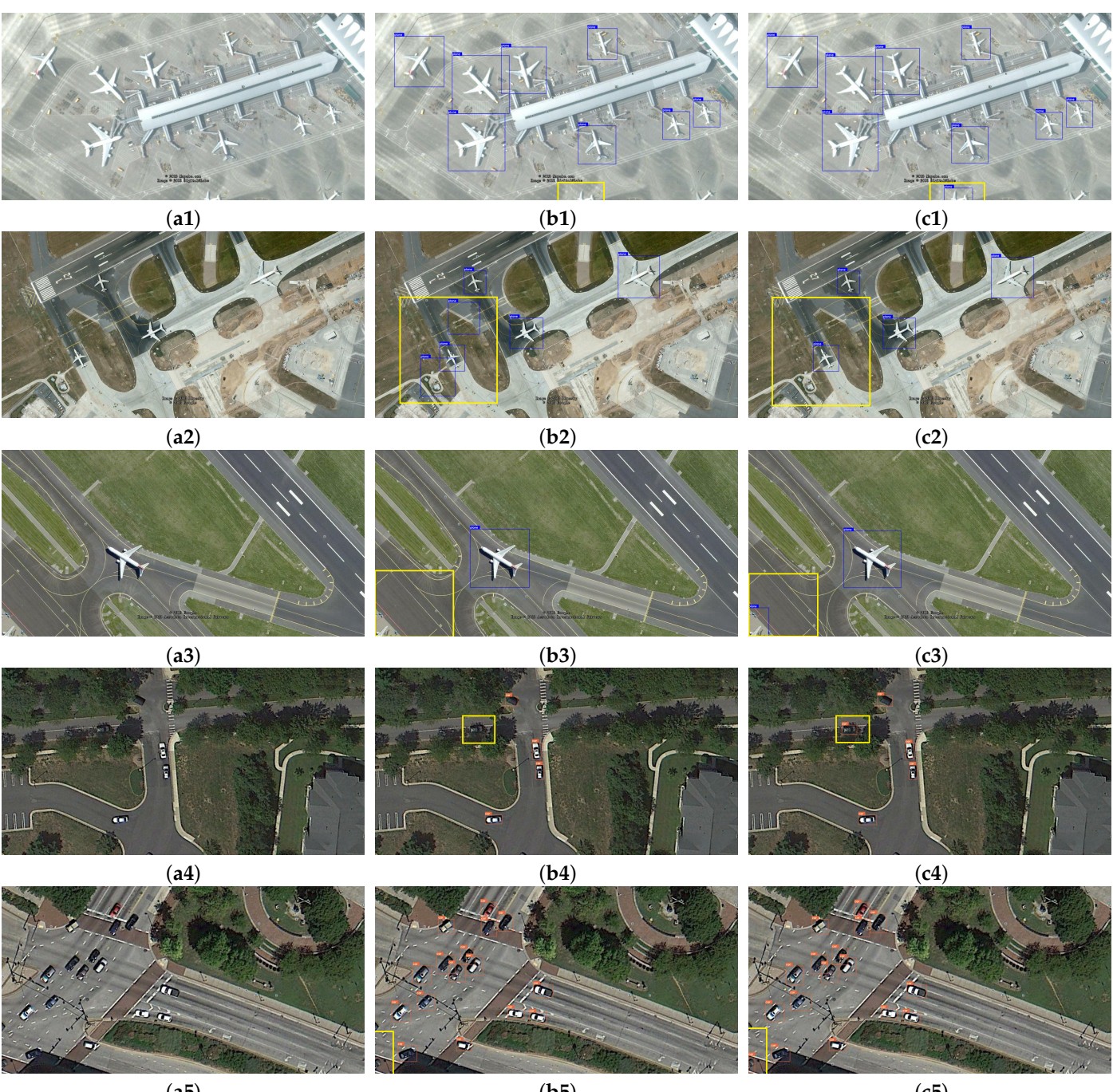

**Figure 10.** *Cont.*

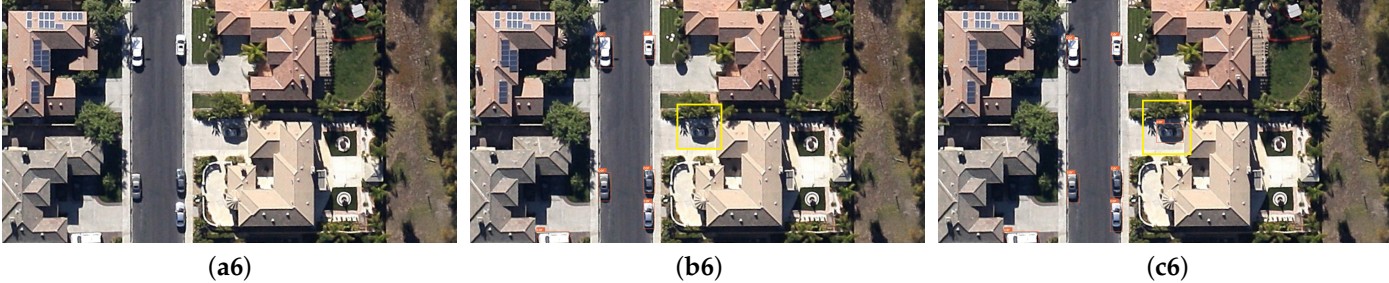

|  (a6)  |  (b6)  |  (c6)  |

**Figure 10.** Comparison of different visualization results on the UCAS-AOD dataset. (**a1**–**a6**) represents the original image; (**b1**–**b6**) represents the visualization results of YOLOv7-Tiny; (**c1**–**c6**) represents the visualization results of the method we proposed. The yellow boxes show the differences between YOLOv7-Tiny and MHLDet.

*3.5. Ablation Study*

To verify the effectiveness of our proposed detector, we conducted ablation studies on the SIMD dataset using the same hyperparameters and parameter settings for each module to ensure a fair and unbiased comparison. We used the precision, recall, mAP, Params, and floating point operations (FLOPs) to verify the availability of the module we proposed. It can be seen in Table 8 and Figure 11 that each of our proposed modules had some improvements over the baseline.

It can be observed from the second and third row of Table 8 that, with the help of SimAM, the mAP improved from 82.3% to 83.6%. Furthermore, with the help of the VNFE, the mAP increased to 83.7%, and the Params and FLOPs decreased by 18% and 10.5%, respectively. This was caused by the absence of semantic information in the backbone before the use of both of them, resulting in less-prominent features. The addition of SimAM enhanced the features of objects to be detected, while VNFE effectively reduced the parameters when extracting the features, which also explains the reason the mAP value improved when the parameters were reduced.

In order to prove the SPPE, we replaced the original SPPCSPC with the SPPE, with the rest of the model remaining the same, from Table 8. In the fourth row, the mAP improved from 82.3% to 83.7%. In the SPPE, we changed the connection mode of the maximum pooling layer and increased its receptive fields from the original 5, 9, and 13 to 5, 13, and 25, so as to enhance the model's detection performance for multi-scale objects.

The validity of the CACL module can be seen from the fifth row in Table 8. The mAP increased from 82.3% to 83.4%. The CACL module effectively utilized the global information and preserved the crucial details in the feature map during downsampling, which led to the enhanced robustness of the model.

**Table 8.** The experimental results of the ablation of each module.

| Method | SimAM | VNFE | SPPE | CACL | mAP | Precision | Recall | Params (m) | FLOPs (G) |
|--------|-------|------|------|------|------|-----------|--------|------------|-----------|
| Baseline | | | | | 82.3 | 77.3 | 79.5 | 6.05 | 13.3 |
| A | ✓ | | | | 83.6 | **84.1** | 78.6 | 6.05 | 13.3 |
| B | | ✓ | | | 83.7 | 76.4 | 82.7 | 4.96 | 11.9 |
| C | | | ✓ | | 83.7 | 76.6 | 82.7 | 6.09 | 13.4 |
| D | | | | ✓ | 83.4 | 77.2 | 82.0 | 6.33 | 13.5 |
| E | ✓ | ✓ | | | 84.1 | 76.3 | **83.8** | 4.96 | 11.9 |
| F | ✓ | ✓ | ✓ | | 84.4 | 77.6 | 82.4 | 5.24 | 12.1 |
| G | ✓ | ✓ | ✓ | ✓ | **84.7** | 77.1 | 82.5 | 5.28 | 12.2 |

Bold text is the optimal value for the column.

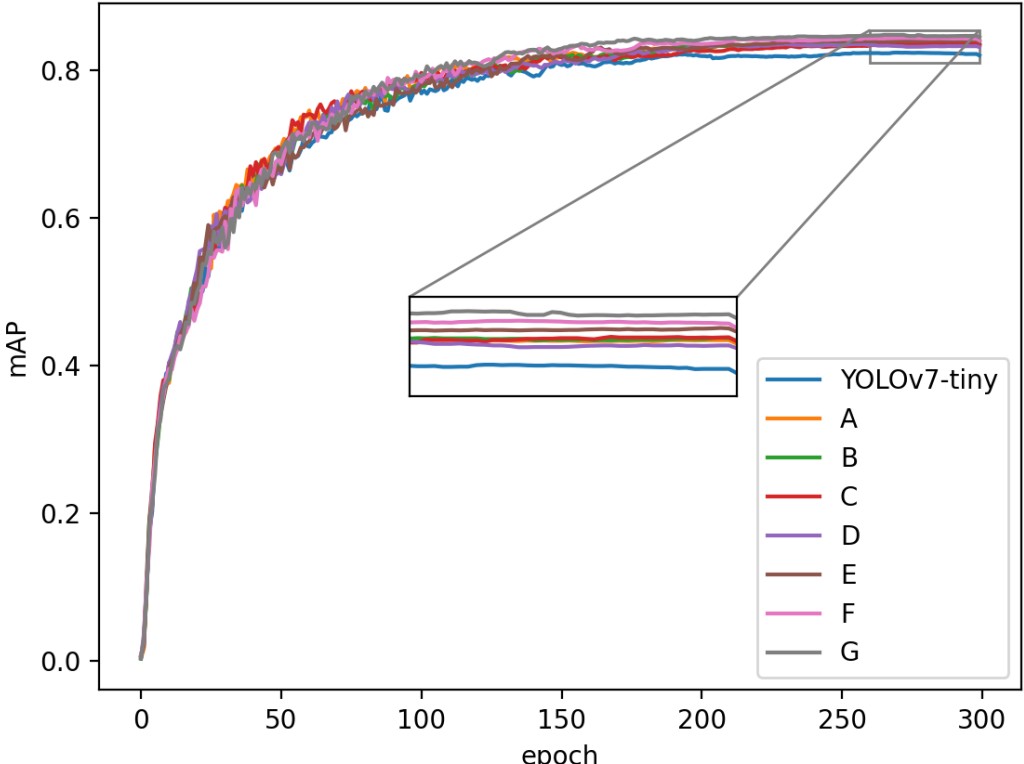

**Figure 11.** Comparison of experimental mAP for ablation. A: SimAM; B: VNFE; C: SPPE; D: CACL; E: SimAM + VNFE; F: SimAM + VNFE + SPPE; G: SimAM + VNFE + SPPE + CACL.

From the sixth and seventh rows of Table 8, it can be observed that stacking multiple modules progressively yielded increasing gains, reaching 84.1% and 84.4%, surpassing the results obtained from using a single module. Additionally, as the achieved mAP continued to rise, the increase in the parameter count also remained within an acceptable range.

Finally, in the eighth row of Table 8, it is evident that the mAP value of applying all the improved modules surpassed that of any individual module alone. It balanced the precision and light weight, enabling higher accuracy even with limited computing resources.

## 4. Discussion

The comparison of the first and eighth rows in Table 8 reveals that, while precision decreased by 0.2%, both the recall and mAP increased by 3% and 2.4%, respectively. This indicates an enhanced ability of the proposed detector to detect objects within remote sensing images, reducing instances of missed detections and facilitating a more-comprehensive recognition of critical features and objects. Furthermore, there was a reduction in the number of parameters and FLOPs by 0.77 m and 1.1 G, respectively. In summary, depending on the specific application and requirements, trading off a slight decrease in precision for an improved recall and mAP can be a justifiable choice.

In addition to the above successes, the proposed detector had certain limitations. MHLDet exhibits suboptimal performance on unmanned aerial vehicle (UAV) images. UAVs can capture images from various poses and viewpoints, leading to variations in the appearance of the same target across different images. This highlights the need for improving the model's generalization ability. Furthermore, when detecting a multitude of rotated objects in RSIs, it can result in horizontal detection boxes containing background regions and overlapping detection boxes, which adversely affects precision. Therefore, the detection performance of MHLDet in this scenario needs to be improved.

## 5. Conclusions

This paper proposes a more-lightweight object detector, MHLDet, based on YOLOv7-Tiny. We integrated SimAM and our proposed VNFE into the backbone network to enhance the model's feature-extraction capability. The improved backbone network also helps the model focus more on critical regions in the feature maps to alleviate the challenge of detecting objects affected by shadows. It can be observed in Figure 9 that MHLDet detects more objects that closely resemble the ground color. Furthermore, the SPPE fuses multi-scale features, enabling the network to handle targets of various scales effectively. Finally, we used the CACL module for downsampling instead of using regular convolution in order to reduce the feature map size and preserve more feature information. The experimental results on the SIMD dataset demonstrated the efficacy of our proposed model. It achieved a reduction in parameters by 12.7% while improving the mAP by 2.4% compared to the baseline and outperformed the majority of the current object-detection models. In our future work, we will focus on dense and rotated objects and enhance the model's detection performance for these objects while maintaining a lightweight design.

**Author Contributions:** Conceptualization, L.Z.; methodology, H.Z.; software, H.Z.; validation, L.Z., H.Z., Z.L. and K.C.; formal analysis, Y.L.; investigation, K.C.; resources, X.Z.; data curation, K.C.; writing—original draft preparation, L.Z. and H.Z.; writing—review and editing, L.Z. and Y.L.; visualization, Z.L.; supervision, Z.L.; project administration, H.Z.; funding acquisition, X.Z. All authors have read and agreed to the published version of the manuscript.

**Funding:** This work was supported by the National Basic Research Program of China (Grant No. 2019YFE0126600); the Major Project of Science and Technology of Henan Province (Grant No. 201400210300); the Key Scientific and Technological Project of Henan Province (Grant No. 212102210496); the Key Research and Promotion Projects of Henan Province (Grant Nos. 212102210393 and 202102110121); the Kaifeng Science and Technology Development Plan (Grant No. 2002001); the National Natural Science Foundation of China (Grant No. 62176087); and the Shenzhen Science and Technology Innovation Commission (SZSTI), the Shenzhen Virtual University Park (SZVUP) Special Fund Project (Grant No. 2021Szvup032).

**Data Availability Statement:** The data used to support the findings of this study are available from the corresponding author upon request.

**Acknowledgments:** We sincerely thank the anonymous Reviewers for the critical comments and suggestions for improving the manuscript.

**Conflicts of Interest:** The authors declare no conflict of interest.

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
