# Peer review of "MHLDet: A Multi-Scale and High-Precision Lightweight Object Detector Based on Large Receptive Field and Attention Mechanism for Remote Sensing Images"

_remotesensing, doi:10.3390/rs15184625_

Round 1
Reviewer 1 Report
You submitted a camera ready version. The paper is suitable for the publication as is.
The quality of English is fine and appropriate.
Author Response
Dear Reviewers:
Thank you for your comments concerning our manuscript entitled “MHLDet: A Multi-Scale and High-Precision Lightweight Object Detector Based on Large Receptive Field and Attention Mechanism” (ID: remotesensing-2492589).
Reviewer 2 Report
No comments!
Author Response

(The authors gave the same response as above.)

Reviewer 3 Report
Aiming at three problems in remote sensing images:1) the significant differences in the size of objects; 2) many objects are hidden in shadows or close to the ground’s color; 3) most detectors have large parameters. This manuscript proposed MHLDet to solve these problems. The intentions of the manuscript are good, but the following questions need to be answered. Finally, a major revision is given.
1. In the manuscript, it is necessary to increase the effectiveness analysis of the proposed method and compare it with other methods. For example, the differences between the existing method and the proposed method are analyzed, as well as the advantages of the proposed method. And so on.
2. The introduction of related work is too brief, and literature search needs to be increased.
3. Some of the descriptions in the manuscript are not rigorous and need to be validated or referenced to increase their persuasiveness. For example, lines 33-36 on page 2.
4. Pay attention to the grammar and formatting in the manuscript. The word is not stated before the abbreviation.
“Finally, Section ?? provides the conclusions from the study.” in line 119.
“CBL” in the manuscript.
“large targets” in line 252
and so on.
Author Response
Dear Reviewers:
Thank you for your comments concerning our manuscript entitled “MHLDet: A Multi-Scale and High-Precision Lightweight Object Detector Based on Large Receptive Field and Attention Mechanism” (ID: remotesensing-2492589). Those comments are all valuable and very helpful for revising and improving our paper, as well as the important guiding significance to our research. We have studied the comments carefully and have made corrections which we hope meet with approval. The main corrections in the paper and the responses to the comments are in the attachment.

Reviewer 4 Report
See the attachment

See the attachment
Author Response

(The authors gave the same response as above.)

Reviewer 5 Report
A multi-scale object detector named MHLDet is proposed in this paper. The experiment shows the lightweight and effective performance of the structure. But I think the model improvement is not novel. And the ablation experiments do not show whether the gain obtained by adding the known module SimAM is better or the gain obtained by the proposed module. There is no correspondence between the problem posed and the way it is solved.
1. The solution to the detector's difficulty in finding objects hidden in shadows in Line88 is not mentioned in the paper. What is the solution to the problem. I think "Large Receptive Field and Attention Mechanism" is added to solve what problems are more worth exploring and more relevant to the title.
2. Figure 1 should show the current problems, and it should get a clearer focus. Currently no focus is expressed.
3. Figure 2 should give a clear view of the focus modules of your model. For example, the location of the SPPE needs to be shown, the Detail section needs to be more clearly expressed rather than using a "=", and the Output section needs to be labeled with what the computational equation represents. Also the title needs to be clearly annotated.
4. Figure 3 and 4 should be annotated with the meaning of each module. Different kernel and stride arrangements I'm sure have more aesthetically pleasing expressions. You can refer to deeplabv for the aspp presentation method.
5. Table 8 needs to show more effects such as SimAM+VNFE, SimeAM+VNFE+SPPE as a way to determine the effectiveness of the proposed modular CACL. The ablation experiments do not clearly show the gain performance of the proposed module.
6. The Conclusions section does not require a detailed description of the experimental procedure. The finding of "Lines 459-461" requires experiments or previous research to prove and the article does not solve the problem, so it is not recommended.
7. More details of the paper need to be noted, e.g. Line119, Section X is displayed in the PDF incorrectly.
1. Sentence structure is disorganized and difficult to understand,Line2-3 " And numerous researchers have conducted extensive research on small objects in RSIs but ignored the significant differences in the size of objects, and there are also plenty of medium and large objects."
2. Sentences are too long, resulting in subjects that can be misinterpreted. This issue needs to be checked and corrected throughout. Lines 29-31 can be split and reorganized appropriately to make them flow better. There is no specific antecedent before which in Line23. The subject in this sentence could be more clearly expressed in Line 143-145.
3. Confusion due to grammatical errors. Line133"YOLOv3-Tiny is a lightweight version of the YOLO series."
Author Response

(The authors gave the same response as above.)

Round 2
Reviewer 3 Report
After the revision, some questions have been answered clearly and the English expression has been improved. However, there are still some small questions left as follows:
The intention and conclusion may be supplemented to Fig.7, and Fig 8, which are more important than the given illustrations.
Pay attention to the grammar and formatting in the manuscript. Such as “Table 1 as the main parameters.” in line 218.
Author Response

(The authors gave the same response as above.)

Reviewer 4 Report
The manuscript can be accepted.
Minor editing
Author Response

(The authors gave the same response as above.)

Reviewer 5 Report
1. Figure and table captions should indicate the dataset from which the results were obtained." Table3,6" "Figure1,7,8".
2. The figure caption should be more accurate. For example, "Figure1" shows the problem of edge detection operation on remote sensing image, not the remote sensing image.
3. The size of the image is also able to be optimized, as shown in Figure 2 for the Overall structure, but it is so large as the internal module Figure3 that it is difficult to see.
1. Line 218“Table 1 as the main parameters”
2. Line 482“resulting in the features is not obvious“
Author Response

(The authors gave the same response as above.)
